# Intraoperative Neuromonitoring Does Not Reduce the Risk of Temporary and Definitive Recurrent Laryngeal Nerve Damage during Thyroid Surgery: A Systematic Review and Meta-Analysis of Endoscopic Findings from 73,325 Nerves at Risk

**DOI:** 10.3390/jpm13101429

**Published:** 2023-09-23

**Authors:** Anna Teresa Cozzi, Alice Ottavi, Paolo Lozza, Alberto Maccari, Roberto Borloni, Letizia Nitro, Elena Giulia Felisati, Andrea Alliata, Barbara Martino, Giancarlo Cacioppo, Manuela Fuccillo, Cecilia Rosso, Carlotta Pipolo, Giovanni Felisati, Loredana De Pasquale, Alberto Maria Saibene

**Affiliations:** 1Otolaryngology Unit, Santi Paolo e Carlo Hospital, Department of Health Sciences, Università Degli Studi di Milano, 20142 Milan, Italy; annateresa.cozzi@unimi.it (A.T.C.); alice.ottavi@unimi.it (A.O.); paolo.lozza@asst-santipaolocarlo.it (P.L.); alberto.maccari@asst-santipaolocarlo.it (A.M.); borlonir@gmail.com (R.B.); letizia.nitro@unimi.it (L.N.); elena.felisati@gmail.com (E.G.F.); andrea.alliata@unimi.it (A.A.); barbara.martino@unimi.it (B.M.); giancarlo.cacioppo@unimi.it (G.C.); manuela.fuccillo@asst-santipaolocarlo.it (M.F.); cecilia.rosso@unimi.it (C.R.); carlotta.pipolo@unimi.it (C.P.); giovanni.felisati@unimi.it (G.F.); 2Thyroid and Parathyroid Surgery Unit, ASST Santi Paolo e Carlo, Department of Health Sciences, Università Degli Studi di Milano, 20142 Milan, Italy; loredana.depasquale@asst-santipaolocarlo.it

**Keywords:** thyroidectomy, vocal folds palsy, lobectomy, hoarseness, adverse event

## Abstract

Background: While intraoperative neuromonitoring (IONM) helps the early identification of recurrent laryngeal nerve (RLN) damage, IONM’s role in RLN damage prevention is not defined, given the lack of large studies on the subject. Methods: In a PRISMA-compliant framework, all original thyroid surgery prospective studies providing early postoperative endoscopic data for all patients were pooled in a random-effects meta-analysis. We compared the temporary (and definitive where available) RLN damage rates according to IONM use and IONM type (intermittent, I-IONM, or continuous, C-IONM). Results: We identified 2358 temporary and 257 definitive RLN injuries in, respectively, 73,325 and 66,476 nerves at risk. The pooled temporary and definitive RLN injury rates were, respectively, 3.15% and 0.422% considering all procedures, 3.29% and 0.409% in cases using IONM, and 3.16% and 0.463 in cases not using IONM. I-IONM and C-IONM, respectively, showed a pooled temporary RLN injury rate of 2.48% and 2.913% and a pooled definitive injury rate of 0.395% and 0.4%. All pooled rates had largely overlapping 95% confidence intervals. Conclusions: Our data suggest that IONM does not affect the temporary or definitive RLN injury rate following thyroidectomy, though its use can be advised in selected cases and for bilateral palsy prevention.

## 1. Introduction

Recurrent laryngeal nerve (RLN) lesions are one of the most common complications that can occur during thyroidectomy. Hoarseness, dysphonia, dysphasia, voice fatigue, and aspiration are well-known outcomes of RLN injuries. The overall incidence of RLN lesions and subsequent vocal cord injuries widely varies among published studies (0.5–20.0%), and recent systematic reviews report a transient incidence of 0.6–9.6% and a permanent incidence of 0.0–2.0% [1].

While intraoperative nerve monitoring (IONM) is capable of aiding and confirming the localization of the recurrent laryngeal nerve (RLN) during various surgical approaches, such as open, endoscopic, or robotic procedures, its widespread adoption has primarily occurred in total thyroidectomies. This is mainly to ensure the intact function of the RLN on one side before proceeding with the contralateral side, thus preventing bilateral vocal cord palsies [2]. Many surgical societies and guidelines recommend the use of IONM, especially in surgery for recurrent cancer, locally advanced cancer, or large goiters, despite partially contrasting literature results [3,4].

Even though IONM use is, at present, widespread, there are few high-evidence studies comparing the rate of RLN injuries between procedures using IONM and without IONM (e.g., only one prospective, randomized case–control evaluation about the efficacy of IONM for voice performance is available, and only in a group robotic surgery subgroup [5]). Therefore, even though we could hypothesize that IONM might prevent RLN injury in thyroid surgery, this matter still remains disputed [6]. Some studies show that IONM reduces the incidence of RLN injuries in high-risk thyroid surgery, such as in oncological surgeries and in patients with previous neck surgery [7,8]. Other research shows that the IONM advantage is limited to preventing double nerve damage and consequent bilateral vocal cord paralysis in total thyroidectomy [9]. In these cases, a so-called loss of signal (LOS) on the first operated side stops the contralateral side dissection.

Given the relatively small incidence of RLN during thyroid surgery, a more definitive analysis of the role of IONM in limiting RLN damage requires both a large patient pool and a careful systematic evaluation of vocal cord function, as the most readily measurable effect of RLN. Therefore, this meta-analysis aims to settle this debate by analyzing a significant bulk of studies that employ a systematic postoperative evaluation of vocal cord function via endoscopic means, comparing the incidence of temporary and definitive RLN injuries with IONM and non-IONM approaches.

## 2. Materials and Methods

This review was registered in the International Prospective Register of Systematic Reviews under the number CRD42022310952.

### 2.1. Search Strategy

A systematic review and meta-analysis were conducted between 2 January 2023 and 4 May 2023, according to the Preferred Reporting Items for Systematic Reviews and Meta-analyses reporting guidelines. We completed systematic electronic searches for studies written in English, Italian, German, French, or Spanish published until the search date that reported rates of vocal cord palsy due to recurrent laryngeal nerve damage during human thyroid surgery and specified whether intraoperative neuromonitoring was used during the procedure.

On 19 January 2023, we searched MEDLINE, Embase, Web of Science, Scopus, Cochrane Library, and ClinicalTrials.gov databases using wide search strategies for thyroid surgery, recurrent laryngeal nerve, and endoscopy or laryngoscopy. The detailed search strategy with the number of unique items retrieved from each database is available in Appendix A.

We included any study dealing with thyroid surgery in humans. We excluded cadaver studies, meta-analyses, systematic and narrative reviews, and case reports, though references from review articles were hand-checked for additional potentially relevant studies. No minimum study population was required. We included only prospective studies that explicitly reported temporary vocal fold palsy rate after surgery (even if nil) and specified using a systematic endoscopic laryngeal examination in all patients, performed between the surgical procedure and postoperative day 7. Furthermore, we included only articles that specified whether or not an IONM device was used and allowed allocating RLN damages to an IONM or non-IONM patient group.

Abstracts and full texts were reviewed in duplicate by two different authors. To maximize the rate of inclusivity in the early stages of the review, at the abstract stage, we included all studies deemed eligible by at least one rater. Then, during the full-text review stage, disagreements were resolved by consensus between raters.

### 2.2. Patient/Population, Intervention, Comparison, Outcomes, Timing, Studies (PICOTS) Criteria

The PICOTS criteria for the present review were as follows:

P: patients undergoing thyroid surgery;

I: thyroidectomy, lobectomy, subtotal thyroidectomy, or completion thyroidectomy;

C: use or no use of intraoperative neuromonitoring;

O: vocal cord palsy due to intraoperative recurrent laryngeal nerve damage identified via endoscopy;

T: events occurring after surgery and identified after awakening from anesthesia and no later than one week after surgery;

S: all prospective original studies except case reports.

### 2.3. Data Extraction

For each included article, we recorded study type, number of surgeries and procedure type (total thyroidectomy, subtotal thyroidectomy, hemithyroidectomy, completion thyroidectomy), access type (open, endoscopic, robotic), number of IONM surgeries and non-IONM surgeries, number of RLNs at risk for each procedure (preferably as stated by authors, otherwise as resulting from number and type of procedures performed), and number of damaged RLNs with and without IONM (temporary and definitive).

Selected studies were assessed for both quality and methodological bias. Randomized clinical trials were rated according to the Cochrane Risk of Bias Tool for Randomized Controlled Trials (CRBT) [10]. All other prospective articles were scored according to the National Heart, Lung, and Blood Institute Study Quality Assessment Tools (NHI-SQAT) [11]. Two authors rated articles in duplicate, with disagreements resolved by consensus. When using the CRBT, items were rated as good, fair, or poor according to the proposed conversion thresholds. When using the NHI-SQAT, items were rated as good if they fulfilled at least 80% of the items required by the NHI-SQAT, fair if they fulfilled between 50% and 80% of the items, and poor if they fulfilled less than 50% of the items, using the methodology we consolidated in prior mixed-level-of-evidence reviews [12,13,14]. Articles rated as being of poor quality according to either score were excluded from the meta-analysis.

Furthermore, among the articles selected for the meta-analysis, those systematically reporting complete data for definitive RLN damages were included in a secondary analysis, again comparing RLN definitive injury rates in IONM and non-IONM groups. Definitive damages had to be defined as such after at least 1 year from the surgical procedure and diagnosed endoscopically. Therefore, we included only studies in which either all patients or patients with temporary damage were systematically followed up endoscopically for at least 1 year after surgery per study design. Furthermore, in this subgroup analysis, we looked for articles reporting the specific type of IONM used (intermittent, I-IONM, or continuous, C-IONM). These articles were used for further analysis comparing the two IONM methods.

Both in the temporary and definitive RLN damage analyses, we included only events that were classified as accidental by the authors. Therefore, cases with planned RLN resection due to oncological reasons were excluded from the data analysis.

The included articles’ levels of evidence were scored according to the Oxford Centre for Evidence-based Medicine (OCEBM) level of evidence guide [15].

### 2.4. Meta-Analysis

The pooled frequency of temporary and definitive RLN damage with 95% confidence intervals was assessed using a random-effects model. Damage rates were compared according to IONM use, again in a random-effects model. The between-study heterogeneity was assessed using Cochran’s Q and I2 statistics. Publication bias was assessed graphically via the funnel plot method and Egger’s and Begg’s tests.

All search results, abstract and article selection, data extraction, and descriptive statistics were performed using the Google Sheets web application (Google LLC, Mountain View, CA, USA). The meta-analysis was performed using Medcalc (version 20.104; MedCalc Software Ltd., Ostend, Belgium).

## 3. Results

### 3.1. Search Results

Among the 3116 unique research items initially identified, 1127 articles were selected to undergo full-text and quality evaluation. Ultimately, 164 studies were retained for further systematic review and meta-analysis (see Figure 1).

The included studies are reported analytically in Appendix A, along with the extracted data.

A total of 80 articles were prospective cohort studies, 72 were prospective case series, and 12 were randomized controlled trials. According to the CRBT and/or NHI-SQAT, 55 articles were rated as low-risk-of-bias/good-quality studies and 109 articles were rated as mild-risk-of-bias/fair-quality studies (16 high-risk-of-bias/poor-quality studies had already been excluded during the full-text analysis phase). Most articles lacked information to support the comparability of patients and, in the case of randomized controlled trials, several issues were found in the blinding of participants and/or evaluators.

The 164 studies reported data on 42,015 procedures and 73,325 nerves at risk. A total of 77 studies used IONM in all patients, 7 studies included procedures performed with and without IONM, and 80 studies did not use IONM in any patient.

For what concerns procedure types, we found 27,561 thyroidectomies, 9719 hemithyroidectomies, 916 subtotal thyroidectomies, and 514 completion thyroidectomies (3305 procedures were not defined, although the articles provided enough data to allow for RLN rate quantification). The included articles included 35,192 open cervicotomy accesses, 3926 endoscopic accesses, and 1759 robotic procedures.

The studies reported 1513 temporary RLN damages in 48,930 nerves at risk for the IONM group, and 846 in 22,936 nerves at risk for the non-IONM group.

Among selected articles, 151 reported complete data on definitive RLN injury rates. In this subgroup, there were 38,269 procedures with 66,476 nerves at risk (48,059 in the IONM group and 17,754 in the non-ION group). There were 257 definitive RLN injuries (171 in the IONM group and 86 in the non-IONM group). These data were included in a further meta-analysis.

Last, 65 articles reported detailed information on the type of IONM used. In this further subgroup, there were 28,187 procedures, with 49,139 RLN at risk (41,261 in the I-IONM group, 7331 in the C-IONM group, and 547 without IONM). There were 1000 temporary and 137 definitive RLN damages in the I-IONM group and 257 temporary and 39 definitive damages in the C-IONM group. These data were used for another meta-analysis subset.

Data on procedures, approaches, and temporary RLN damage rates are reported in Appendix A. Data on definitive RLN damage rates are reported in Appendix A. Data on temporary and definitive RLN damage according to IONM type are reported in Appendix A.

### 3.2. Meta-Analysis

#### 3.2.1. Temporary RLN Damage

When considering all RLNs at risk during thyroid surgery, the pooled RLN temporary damage rate was 3.15% (95% CI, 2.73–3.59%). While the funnel plot method suggested some degree of publication bias (see Appendix A), Egger’s and Begg’s tests were negative, with respective *p*-values of 0.1249 and 0.9433. There was significant heterogeneity in the gathered data (I2 88.93%, *p* < 0.0001). Appendix A reports the forest plot for all included studies.

Results were roughly similar when distinguishing procedures performed with and without IONM. For IONM procedures, the pooled temporary damage was 3.29% (95% CI, 2.69–3.95%). While the funnel plot method suggested some degree of publication bias (see Appendix A), Egger’s and Begg’s tests were negative, with respective *p*-values of 0.1249 and 0.9433. There was significant heterogeneity in the gathered data (I2 91.37%, *p* < 0.0001). Appendix A reports the forest plot for IONM studies. For non-IONM procedures, the pooled temporary damage was 3.16% (95% CI, 2.54–3.86%). While the funnel plot method suggested some degree of publication bias (see Appendix A), Egger’s and Begg’s tests were negative, with respective *p*-values of 0.9469 and 0.9795. There was significant heterogeneity in the gathered data (I2 86.47%, *p* < 0.0001). Appendix A reports the forest plot for non-IONM studies.

#### 3.2.2. Definitive RLN Damage

When considering all RLNs at risk during thyroid surgery, the pooled RLN definitive damage rate was 0.422% (95% CI, 0.341–0.513%). While the funnel plot method did not show any degree of publication bias (see Appendix A), Egger’s and Begg’s tests were positive, with respective *p*-values of 0.0021 and <0.0001. There was some heterogeneity in the gathered data (I2 49.76%, *p* < 0.0001). Appendix A reports the forest plot for all included studies.

Results were roughly similar when distinguishing procedures performed with and without IONM. For IONM procedures, the pooled definitive damage was 0.409% (95% CI, 0.302–0.532%). While the funnel plot method did not show any degree of publication bias (see Appendix A), Egger’s and Begg’s tests were positive, with respective *p*-values of 0.0029 and 0.001. There was some heterogeneity in the gathered data (I2 59.76%, *p* < 0.0001). Appendix A reports the forest plot for IONM procedures. For non-IONM procedures, the pooled definitive damage was 0.463% (95% CI, 0.339–0.607%). While the funnel plot method did not show any degree of publication bias (see Appendix A) and Egger’s test was negative (*p* = 0.6187), Begg’s test was positive with a *p*-value <0.0001. There was some heterogeneity in the gathered data (I2 49.76%, *p* = 0.0081). Appendix A reports the forest plot for non-IONM procedures.

#### 3.2.3. Type of IONM

When comparing IONM types, two different analyses were performed on each type of group, one for temporary damage and one for definitive damage.

For I-IONM, the pooled temporary damage rate was 2.48% (95% CI, 1.868–3.176%). The funnel plot method did not show any degree of publication bias (see Appendix A), and Egger’s and Begg’s tests were negative, with respective *p*-values of 0.3013 and 0.9466. There was substantial heterogeneity in the gathered data (I2 92.45%, *p* < 0.0001). Appendix A reports the forest plot for temporary damage in I-IONM procedures. For I-IONM, the pooled definitive damage rate was 0.395% (95% CI, 0.277–0.534%). The funnel plot method did not show any degree of publication bias (see Appendix A), while Egger’s and Begg’s tests were positive, with respective *p*-values of 0.0255 and 0.028. There was some heterogeneity in the gathered data (I2 59.83%, *p* < 0.0001). Appendix A reports the forest plot for definitive damage in I-IONM procedures.

For C-IONM, the pooled temporary damage rate was 2.913% (95% CI, 1.728–4.395%). The funnel plot method did not show any degree of publication bias (see Appendix A). Egger’s test was negative (*p* = 0.1518) and Begg’s test was positive (*p* = 0.0279). There was substantial heterogeneity in the gathered data (I2 92.55%, *p* < 0.0001). Appendix A reports the forest plot for temporary damage in C-IONM procedures. For C-IONM, the pooled definitive damage rate was 0.4% (95% CI, 0.138–0.798%). The funnel plot method did not show any degree of publication bias (see Appendix A), while Egger’s and Begg’s tests were positive, with respective *p*-values of 0.0255 and 0.028. There was some heterogeneity in the gathered data (I2 81.56%, *p* < 0.0001). Appendix A reports the forest plot for definitive damage in C-IONM procedures.

## 4. Discussion

To the authors’ knowledge, no other systematic reviews specifically address the risk of RLN damage during thyroid surgery with and without IONM in a wide cohort of patients. Our study is further strengthened by including only studies employing systematic endoscopic evaluation of vocal fold movement after surgery. Our study shows that thyroid surgery procedures employing IONM hold the same RLN injury rate as procedures performed without IONM, both in the temporary (3.29% for IONM and 3.16% for non-IONM) and definitive setting (0.409% for ION and 0.463 for non-IONM).

Other studies have been published on this subject, though with significantly smaller data pool sizes. In a meta-analysis, Ku et al. [16] included 3040 patients undergoing thyroidectomies with IONM; they found that the proportion of nerves at risk (NAR) with temporary RLN paralysis post operation was 2.26% (95% CI: 1.6–2.9, I2 = 37), similar to our result, while the proportion of NAR with permanent RLN palsy post operation was 0.05% (95% CI: 0.08–0.2, I2 = 0). However, this study did not explore RLN paralysis rate without IONM. In a more recent meta-analysis, Davey et al. included eight RCTs reporting data on 2521 patients, with 49.8% undergoing IONM procedures (2480/4978) and 50.2% undergoing RLN visualization alone (VA) (2497/4978) [17]. Overall RLN injuries rates were higher for VA (VA: 3.2% (80/2497) vs. IONM: 2.3% (58/2480)). Permanent RLN injury rates were slightly higher for VA (VA: 0.6%, (12/2497) vs. IONM: 0.5%, (12/2480), OR: 0.76, 95% CI: 0.36–1.59, P = 0.470, I2 = 0%). They concluded that, compared to VA alone, using IONM failed to significantly reduce RLN injury rates during thyroid surgery. The slight difference in RLN injuries with IONM from our results is with all due probability due to the different sample sizes, as in other studies [7,18]. Other meta-analyses described similar results, although only a fraction of the included studies systematically employed postoperative laryngoscopy to assess vocal cord damage, thus potentially underestimating the incidence of RLN injury [19]. Other meta-analyses, on the other hand, demonstrate the merit of IONM in preventing transient injury during thyroidectomy, but these results are not so predictable, due to the only clinical dysphonia evaluation [20]. Though other meta-analyses exploring the role of IONM in RLN damage do exist [17,21,22], our study is the only study dealing with an extremely significant patient pool, thus allowing for appreciating slight differences even in rare events and the only one that specifically relies on prospective studies that explicitly and systematically employ laryngeal endoscopy for evaluation. We can assume that our data, albeit heterogeneous, allows us to obtain solid statistical results and offer an innovative comparison.

In these regards, it is interesting to observe that we have not found any significant difference not only between procedures using and not using IONM but also when comparing the two main types of IONM, i.e., C-IONM and I-IONM. It appears that the 3% and 0.4% rates—respectively, for temporary and definitive RLN damages—should be considered somehow “endemic” to thyroid surgery. There are indeed several factors that might affect RLN injury risks, such as the procedure volume at a specific medical center, the surgeon’s experience, or the specific histology of the disease being treated, regardless of the specific neuromonitoring technique being used, and that must be taken into account when evaluating each specific case. Consequently, the lack of a general protective role for IONM toward RLN injuries we showed in this work does not diminish the need for identifying the specific situations where this device does improve the patient and surgeon experience. As a prime example, it has been shown that IONM has a definite role in training, where its use allows for safe procedures comparably to the presence of an expert mentor [23].

It is indeed true that our analysis has more than a few limits. The most prominent includes both RCT and other prospective studies. Although this choice reduces the overall level of evidence from a formal standpoint, it is indeed true that the 12 RCTs identified [5,18,24,25,26,27,28,29,30,31,32,33] would not offer solid evidence for such a rare occurrence. Indeed, the aim of this study was to achieve maximum significance and reduce our confidence intervals by expanding the number of nerves at risk.

However, including only studies with a postoperative evaluation of glottic mobility determines a significant loss rate, although improving our results. On a side note, using glottic mobility as an indirect measure of RLN function introduces a potential analytic bias, although there are at present no other comparably widely and conveniently available instruments to measure postoperative RLN function.

Moreover, the results were not evaluated separately in different surgery approaches (open, robotic, endoscopic). Again, there was significant heterogeneity in terms of age, sex, previous surgeries, histologic diagnosis, neck dissection, and previous local radiotherapy treatments in the overall cohort of patients. This heterogeneity is well documented through Egger’s and Begg’s tests performed at the meta-analysis stage. Despite subgrouping the patients according to the IONM use and the type of IONM, this heterogeneity remains constant, thus underlying the strong role of other patient and disease characteristics. Even if these patient- and disease-specific parameters do have a role in the RLN injury rate and might have affected single studies’ results, our study nevertheless aimed at defining the baseline risk. Our analysis aimed to look at a relatively infrequent injury event over a significant pool of patients in order to offer specific guidance when offering thyroid surgery to patients, without analyzing specific techniques. Such a detailed analysis of injury determinants was therefore outside the scope of our work and might be the subject of a further meta-analysis. Despite the good evidence level and methodological consistency of our meta-analysis, it is evident that more RCTs with fewer heterogeneous features and selective inclusion criteria are needed to assess the role of IONM in specific patient populations.

It has to be noted that this review does not cover the role of IONM in identifying and sparing the external branch of the superior laryngeal nerve, which plays a significant role in voice outcomes. IONM has been shown to potentially play an important role in identifying and sparing this structure, whose damage cannot be evaluated endoscopically [34].

## 5. Conclusions

In conclusion, our results, based on one of the largest endoscopy-evaluated patient pool of the present literature, does not undermine the IONM importance in avoiding bilateral RLN damage in thyroid surgery. It appears though that in the general population, IONM, regardless of the technique used, does not play a significant role in preventing RLN damage. Such a lack of influence on the RLN injury rate strengthens the concept that IONM might be employed at surgeons’ discretion and should not be considered a must-have tool for performing thyroid surgery, even from a legal standpoint. Nevertheless, our review does not cover specific complex scenarios (such as reinterventions) where IONM can play a decisive role, nor does it delve into the significance of these tools for training and teaching purposes. Defining these essential points of IONM usage is in our opinion extremely relevant, given the call for more information on this tool advocated by otolaryngology residents [35]. IONM remains a basic tool for thyroid surgery and further research might help define which scenarios are positively and significantly influenced by its use.

## Figures and Tables

**Figure 1 jpm-13-01429-f001:**
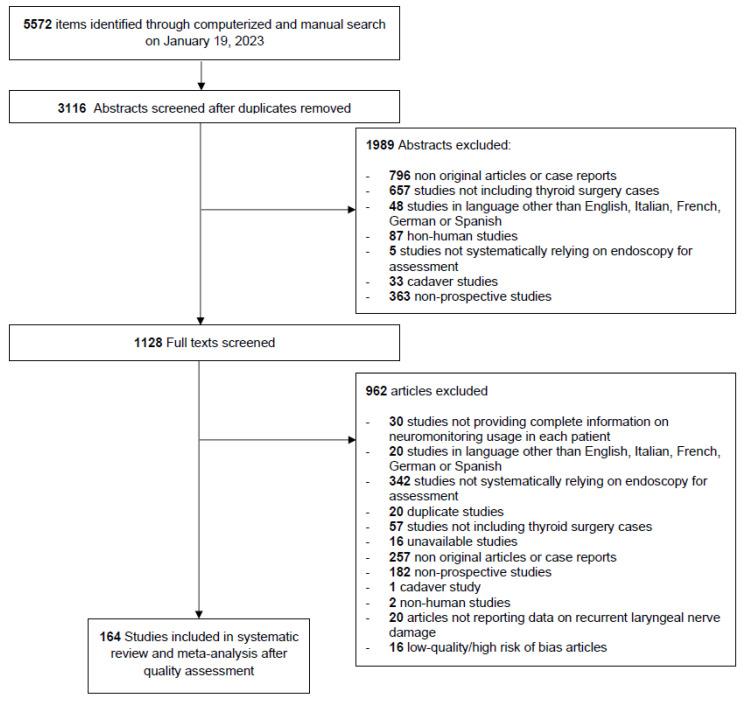
PRISMA-style flowchart of article selection through the systematic review and meta-analysis process.

## Data Availability

All data pertaining to this meta-analysis are available from the authors upon reasonable request.

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
