# Peer review of "Intraoperative Neuromonitoring Does Not Reduce the Risk of Temporary and Definitive Recurrent Laryngeal Nerve Damage during Thyroid Surgery: A Systematic Review and Meta-Analysis of Endoscopic Findings from 73,325 Nerves at Risk"

_jpm, 2023, doi:10.3390/jpm13101429_

Round 1

Reviewer 1 Report

I read with interest the systematic review and meta-analysis of Cozzi et Al. on the absence of reduction of temporary and definitive recurrent laryngeal nerve (RLN) palsy using intraoperative neuromonitoring.

RLN injuries represent one of the most feared complications after thyroid and parathyroid surgery inducing a significant postoperative morbidity.

The difficulty of dissecting and finding the RLN during cervical surgery also lies in the great anatomic variability of its position and sometimes is due to an early division in branches but with only one motor branch that is the only functional and usually located anteriorly.

The use of intraoperative neurostimulator is widely used to detect this motor branch (distinguishing it from the sensitives branch/es) and confirm its function.

The author in this article concludes that IONM doesn’t affect temporary or definitive RLN injury rate following thyroidectomy and that IONM use should be at surgeons’ discretion and should not be considered a must-have tool for performing thyroid surgery.

Even if I completely disagree with the conclusions of the authors the literature review is well conducted, and results seems convincing not finding a clear superiority of IONM use on the reduction of RLN injury rate.

Personally, I think that the conclusion should be reformulated since we are not facing an absolute truth but just a literature review with all its implicit limits.

Moreover, admitting that IONM use doesn’t affect the reduction of RLN injury rate there are many other advantages (someone also mentioned by the authors) that alone could justify the systematic use of the IONM.

Some comments:

Please avoid statements like "this is the only review" as in lines 210-217  because reduces the scientific sound of the article.

Please define in the text transient and definitive RLN palsy: 6 or 12 months for definitive?  

Since transient injuries have different recovery times (usually between 4 and 6 weeks to complete recovery), up to 12 months in line 84-85 how could you evaluate transient/definitive RLN palsy if you have included only patients with laryngeal examination from surgery to only day 7? Which was the follow up in the different articles?  

Another important point: how many patients had the intentional resection of the RLN for oncological reasons in the various selected articles? Have they been included or not? If yes, this would create a bias on the results.

Would be interesting comparing also the effect with the use of Continuous-IONM, could you add and differentiate the results?

Best regards.

Author Response

Dear Editors,

we submit to your kind attention the revised version of our paper “Intraoperative Neuromonitoring doesn’t Reduce the Risk of Temporary and Definitive Recurrent Laryngeal Nerve Damage during Thyroid Surgery. A Systematic Review and Meta-Analysis of Endoscopic Findings from 73,325 Nerves at Risk”.

We’d like to thank you for your prompt and complete review of our paper.

Following the reviewers’ kind advice we’ve performed the requested revisions of the paper. Each criticism reported is addressed next for the sake of clarity, highlighted in yellow in the revised version of the paper. Furthermore, we amended the length and reference issues pointed out by the Assistant Editor.

Reviewer #1:

I read with interest the systematic review and meta-analysis of Cozzi et Al. on the absence of reduction of temporary and definitive recurrent laryngeal nerve (RLN) palsy using intraoperative neuromonitoring. RLN injuries represent one of the most feared complications after thyroid and parathyroid surgery inducing a significant postoperative morbidity. The difficulty of dissecting and finding the RLN during cervical surgery also lies in the great anatomic variability of its position and sometimes is due to an early division in branches but with only one motor branch that is the only functional and usually located anteriorly. The use of intraoperative neurostimulator is widely used to detect this motor branch (distinguishing it from the sensitives branch/es) and confirm its function. The author in this article concludes that IONM doesn’t affect temporary or definitive RLN injury rate following thyroidectomy and that IONM use should be at surgeons’ discretion and should not be considered a must-have tool for performing thyroid surgery. Even if I completely disagree with the conclusions of the authors the literature review is well conducted, and results seems convincing not finding a clear superiority of IONM use on the reduction of RLN injury rate.

--Thank you for appreciating our study and methods despite and honestly reporting your personal position

Personally, I think that the conclusion should be reformulated since we are not facing an absolute truth but just a literature review with all its implicit limits. Moreover, admitting that IONM use doesn’t affect the reduction of RLN injury rate there are many other advantages (someone also mentioned by the authors) that alone could justify the systematic use of the IONM.

--We agree that our conclusions (both in the abstract and in the text) might have sounded a bit too harsh. We reformulated our statements, softened them, and explored a bit more on the other useful roles that IONM has in thyroid surgery, as we agree that it should represent an important tool for any surgeon.

Please avoid statements like "this is the only review" as in lines 210-217 because reduces the scientific sound of the article.

--Thank you for pointing out. This has been amended in the text.

Please define in the text transient and definitive RLN palsy: 6 or 12 months for definitive?

--12 months, as stated below. This threshold has been made explicit in the article.

Since transient injuries have different recovery times (usually between 4 and 6 weeks to complete recovery), up to 12 months in line 84-85 how could you evaluate transient/definitive RLN palsy if you have included only patients with laryngeal examination from surgery to only day 7? Which was the follow-up in the different articles?

--Indeed these points are important and had not been explicit in the first draft of the article. As it’s now reported in the paper, for the definitive analysis we included only studies where patients were followed up endoscopically and systematically for at least 1 year.

Another important point: how many patients had the intentional resection of the RLN for oncological reasons in the various selected articles? Have they been included or not? If yes, this would create a bias on the results.

--The events included in the meta-analysis are only those that were defined as accidental by the authors, planned resections were excluded by the analysis

Would be interesting comparing also the effect with the use of Continuous-IONM, could you add and differentiate the results?

--We added a new subset of meta-analysis addressing this issue, abstract, materials and methods, results, and discussion have been amended consequently. Though the pooled results were overlapping between C-IONM and I-IONM, performing this subset of analysis, seems interesting and pertinent, so thank you for the idea!

Sincerely hoping that you shall find the revision complete and the article worthy of publication in your journal,

Looking forward to hearing from you

Kindest regards,

The authors

Reviewer 2 Report

The topic is exceptionally fascinating, representing the most comprehensive study of neuromonitoring's application in thyroid surgery. While neuromonitoring may not be an absolute necessity for thyroid surgery, its absence could hinder the advancement of surgical techniques in this field. Therefore, it is highly recommended for surgeons to incorporate it into their educational journey.

I observed a minor discrepancy in Figure 1, where the numbers in the last three panels have been miscalculated, leading to inaccuracies in the text between lines 146-152.

Please change the sentence: 
"Although intraoperative nerve monitoring (IONM) can facilitate and confirm the identification of the recurrent laryngeal nerve (RLN), and it can be used in each kind of surgical approach, open, endoscopic, or robotic, its use gained popularity mostly in total thyroidectomies to confirm single side RLN function before completing the procedure on the contralateral side, as to avoid bilateral vocal cord palsies."

My suggestion is:

"While intraoperative nerve monitoring (IONM) is capable of aiding and confirming the localization of the recurrent laryngeal nerve (RLN) during various surgical approaches, such as open, endoscopic, or robotic procedures, its widespread adoption has primarily occurred in total thyroidectomies. This is mainly to ensure the intact function of the RLN on one side before proceeding with the contralateral side, thus preventing bilateral vocal cord palsies."

Author Response

Dear Editors,

We submit to your kind attention the revised version of our paper “Intraoperative Neuromonitoring doesn’t Reduce the Risk of Temporary and Definitive Recurrent Laryngeal Nerve Damage during Thyroid Surgery. A Systematic Review and Meta-Analysis of Endoscopic Findings from 73,325 Nerves at Risk”.

We’d like to thank you for your prompt and complete review of our paper.

Following the reviewers’ kind advice we’ve performed the requested revisions of the paper. Each criticism reported is addressed next for the sake of clarity, highlighted in yellow in the revised version of the paper. Furthermore, we amended the length and reference issues pointed out by the Assistant Editor.

Reviewer #2

The topic is exceptionally fascinating, representing the most comprehensive study of neuromonitoring's application in thyroid surgery. While neuromonitoring may not be an absolute necessity for thyroid surgery, its absence could hinder the advancement of surgical techniques in this field. Therefore, it is highly recommended for surgeons to incorporate it into their educational journey.

-Thank you for your comments. I made anyway clearer in the manuscript that we do support the use of neuromonitoring, as it has many advantages besides its role in RLN damages.

I observed a minor discrepancy in Figure 1, where the numbers in the last three panels have been miscalculated, leading to inaccuracies in the text between lines 146-152.

-Thank you for pointing out! I reported the correct numbers in the article but I made some mistakes in reporting them in the Figure, which has now been amended.

Please change the sentence:

"Although intraoperative nerve monitoring (IONM) can facilitate and confirm the identification of the recurrent laryngeal nerve (RLN), and it can be used in each kind of surgical approach, open, endoscopic, or robotic, its use gained popularity mostly in total thyroidectomies to confirm single side RLN function before completing the procedure on the contralateral side, as to avoid bilateral vocal cord palsies." My suggestion is: "While intraoperative nerve monitoring (IONM) is capable of aiding and confirming the localization of the recurrent laryngeal nerve (RLN) during various surgical approaches, such as open, endoscopic, or robotic procedures, its widespread adoption has primarily occurred in total thyroidectomies. This is mainly to ensure the intact function of the RLN on one side before proceeding with the contralateral side, thus preventing bilateral vocal cord palsies."

-Thank you for your suggestion, we incorporated it in the new version of our manuscript.

Sincerely hoping that you shall find the revision complete and the article worthy of publication in your journal,

Looking forward to hearing from you

Kindest regards,

The authors

Round 2

Reviewer 1 Report

Thank you for the revision no more comments

Reviewer 2 Report

I have reviewed the revised version of your article, and I am pleased to inform you that I accept it in its current form. The revisions have significantly improved the quality and clarity of the content. Thank you for your diligence and for incorporating the suggested changes.